# A Novel Hybrid Membrane Process Coupled with Freeze Concentration for Phosphorus Recovery from Cheese Whey

**DOI:** 10.3390/membranes13040450

**Published:** 2023-04-21

**Authors:** Ipan Hidayat, Lidia Paredes, Pablo M. Binder, Nagore Guerra-Gorostegi, Mabel Mora, Sergio Ponsá, Darren L. Oatley-Radcliffe, Laia Llenas

**Affiliations:** 1BETA Technological Center (TECNIO Network), University of Vic-Central University of Catalonia (UVic-UCC), Carretera de Roda 70, 08500 Vic, Spain; 2Energy Safety Research Institute (ESRI), College of Engineering, Swansea University, Bay Campus, Swansea, Wales SA1 8EN, UK

**Keywords:** cheese whey, freeze concentration, membrane system, microfiltration, phosphorus, ultrafiltration

## Abstract

The ever-increasing demand for phosphorus fertilisers for securing global food production, coupled with finite phosphate rock reserves, is one of the emerging problems in the world. Indeed, phosphate rock is listed as an EU critical raw material, triggering attention to find an alternative source to substitute the use of this limited resource. Cheese whey, characterized by a high content of organic matter and phosphorus, represents a promising feedstock for phosphorus recovery and recycling. An innovative application of a membrane system coupled with freeze concentration was assessed to recover phosphorus from cheese whey. The performances of a microfiltration membrane (0.2 µm) and an ultrafiltration (200 kDa) membrane were evaluated and optimized under different transmembrane pressures and crossflow velocities. Once the optimal operating conditions were determined, a pre-treatment including lactic acid acidification and centrifugation was applied to increase the permeate recovery. Finally, the efficiency of progressive freeze concentration for the treatment of the permeate obtained from the optimum conditions (UF 200 kDa with TMP of 3 bar, CFV of 1 m/s and lactic acid acidification) was evaluated at specific operating conditions (−5 °C and 600 rpm of stirring speed). Finally, 70% of phosphorus could be recovered from cheese whey using the coupled technology of the membrane system and freeze concentration. A phosphorus-rich product was obtained with high agronomic value, which constitutes a further step towards establishing a broader circular economy framework.

## 1. Introduction

Phosphorus (P) is an essential macronutrient for plant growth and one of the most limiting nutrients in guaranteeing crop yield required for securing food availability for the exponential growth of the global population [1]. Phosphorous rock (PR) is a finite resource and is currently the major feed stock for the raw element used to produce P fertilisers [2]. Estimations suggest that PR resources will be completely consumed in 30 to 300 years [3,4]. Nonetheless, along with the application of P fertiliser, nutrient run-offs and excess load may lead to eutrophication of water bodies [5,6]. Furthermore, the economic and geopolitical sectors have led to the concern of P shortage within the European Union (EU), considering that less than 1% of mining sites are located in the EU and 74% of the world’s remaining reserves are controlled by Morocco [7,8]. Therefore, the European Commission declared PR as one of the twenty critical raw materials in 2014 [9]. According to the European Commission’s Circular Economy Action Plan along with the new fertilizer regulations (2019/1009), there is a crucial need to explore alternative secondary raw material-based fertilisers from biogenic waste to recover and recycle P and reduce dependency on imported PR.

The dairy processing industry has recently been considered the EU’s most prominent contributor of industrial food wastewater generation [10]. Cheese is the most popular dairy product, which is projected to produce 1.04 million metric tonnes within the EU, and represents 20% of the world’s total milk production [11,12]. Cheese manufacturing generates a large amount of cheese whey (CW), which represents higher organic and nutrient contents, such as P in the range of 0.1 to 0.6 g/L [13,14] and nitrogen (N) in the range of 0.2 to 2 g/L [15,16] compared to other sources of wastewater, for example, winery, olive mills, and aquaculture wastewater [17]. For that reason, CW signifies as a potential and promising secondary source to be exploited for nutrient recovery, especially P [18,19].

Over the last decades, a variety of physical [11,20,21], biological [10,22,23], and chemical [24,25] treatments have been proposed to treat CW; however, to the authors best knowledge, P recovery has not been fully addressed from CW. Therefore, a hybrid technology coupling membrane technology and freeze concentration (FC) has been assessed with a focus on recovering P.

Pressure-driven membrane separation technology, such as microfiltration (MF) and ultrafiltration (UF), are widely used in the dairy industry [26]. They offer some advantages over the conventional treatments, including low-temperature operations, no phase transitions, high permeate flux, simple procedure, and ease to scale up [27,28,29]. Up until now, membrane fouling due to the accumulation of whey protein or protein-aggregates such as β-lactoglobulin (β-Lg), bovine serum albumin, and α-lactalbumin (α-La) on the membrane surface has been the central problem in the application of pressure-driven membranes to the dairy sector [30,31]. Fouling results in flux decline, which limits the process efficiency and increases the cleaning cost [32,33]. Several pre-treatments have been proposed to boost the MF and UF efficiency and increase the permeate flux, such as the removal of lipids by the precipitation process at pH 4.6 [34], calcium or calcium phosphate precipitation [35], chemical pre-treatment by NaCl addition [33], and the defatting–clarifying process [31]. Furthermore, permeate flux can be improved by modifying the operating conditions, such as increasing crossflow velocity (CFV) and transmembrane pressure (TMP). However, limited studies have attempted to investigate the effects of pre-treatment, such as acidification with lactic acid to pH 3 and centrifugation to produce clarified supernatants, and the resulting decrease that would be expected on fouling behaviour.

FC is a technique for concentrating solutions by freezing the water content and removing the so-formed ice crystals, which eliminates solutes or impurities [36,37]. Compared to other concentration technologies, such as evaporation, the FC process preserves thermally fragile components and requires less energy (0.33 kJ/g water) than evaporation (2.26 kJ/g) [38,39,40,41]. One of the indirect freezing methods available is progressive freeze concentration (PFC), which has been confirmed as a cost-effective and efficient method in the dairy industry [37,42,43,44]. Separation of a single large ice crystal from the concentrated solution is relatively simple and no additional heat exchanger is required, thus substantially reducing the operational cost [40,45]. Several previous studies have investigated the use of FC in the dairy industries. Canella et al. [46] concentrated CW for fermented lactic beverages, and Sánchez et al. [19] concentrated lactose and protein from whey. Moreover, Belén et al. [47] applied one-stage falling-film FC to recover nitrogen, protein, and lactose from CW with an efficiency of 60%, 63%, and 46%, respectively. This technique is indeed a potential promising procedure due to the added recovery of high-quality water, concentrating up to 40–50% (*w*/*w*), operation at low temperature, and low energy consumption [41,48]. Furthermore, Uald-Lamkaddam et al. [49] applied progressive FC to recover protein and lactose from CW; the results showed that 13.9% (*w*/*w*) was recovered in the concentrated fraction, meanwhile 6.2% (*w*/*w*) of the lactose could be recovered in the ice fraction.

Taking into consideration this lacking information, the core objective of this present study was to evaluate the phosphorus recovery efficiency from CW with a hybrid technology of membranes coupled with FC. To reach this objective, the focuses of the work were (1) to optimize the application of different type of membranes under various TMP and CFV; (2) to evaluate the effects of the pre-treatment process by lactic acid acidification and centrifugation on permeate rate and P recovery efficiency; (3) to evaluate the efficiency of FC in terms of P recovery at the temperature of −5 °C and with 600 rpm stirring speed.; and finally (4) to study the potential two-stage membrane filtration system coupled with FC.

## 2. Materials and Methods

### 2.1. Cheese Whey

The raw goat CW was collected from a local cheese-processing factory in Barcelona (Spain). The CW was immediately transported to the laboratory, homogenised, and kept in the fridge (4 °C). The physicochemical characteristics of CW (Table 1) were analysed weekly in triplicate immediately after sampling until the end of the study.

### 2.2. Experimental Setup

A commercial bench-scale crossflow filtration unit, a stainless steel SEPA CF Membrane Element Cell (Sterlitech Corporation, Kent, WA, USA), was used to conduct all experiments in this study (Figure 1 and Appendix A). The SEPA CF Membrane Cell is designed to handle a maximum pressure of 69 bars and to accommodate 19.1 cm × 14 cm of flat sheet membrane.

Furthermore, the experimental system was configured with the following parts: (1) a high-pressure feed pump (M-03 model, Wanner Engineering Inc., Minneapolis, MN, USA); (2) 5.7 L of stainless-steel conical feed tank (Sterlitech Corporation, Kent, WA, USA) to accommodate the feed and retentate returns from the system; (3) a high pressure relief valve (Sterlitech Corporation, Kent, WA, USA); (4) a by-pass needle valve (SS-1RS6, Swagelok, Camarillo, CA, USA) to control the flow of the feed water; (5) two pressure gauges (0–10 bar, Swagelok, Camarillo, CA, USA), placed in both feed and retentate lines, to monitor the water pressure in both lines; (6) two needle valves (Para Plate Corporation, Auburn, CA, USA) to regulate low (0–27 bar) and high (27–69 bar) pressure in the retentate line; (7) a hydraulic hand pump (SPX Power Team, Rockford, IL, USA) to maintain the hydraulic pressure inside the cell holder; and (8) a site read panel mount flow meter (F-550, Blue-White industries Ltd., Huntington Beach, CA, USA) to measure the flow of the retentate.

A flat sheet microfiltration (MF) and ultrafiltration (UF) membranes (Synder, Vacaville, CA, USA) with 0.2 µm and 200 kDa pore sizes, respectively, were used for the study. Both membranes were made of polyvinylidene fluoride (PVDF) with an active area of 140 cm^2^.

### 2.3. Filtration Conditions

#### 2.3.1. Membrane Precondition

Prior to each experiment, pre-conditioning of the membrane was conducted with deionised water and the selected MF 0.2 µm and UF 200 kDa membranes under different TMPs and CFVs (Table 2) until reaching the stable flux rate (MF = 281 L/h·m^2^ and UF = 230 L/h·m^2^) at the room temperature (20 °C). The TMPs and CFVs for the membrane pre-conditioning should be maintained at similar values to the membrane filtration with CW to avoid a change in filtration condition. The main aim when pre-conditioning the membrane is to avoid any change in the physical properties of the membrane during the experiment run. Furthermore, this process also removes any preservatives or other residuals from the membrane and consequently avoids permeate decline by membrane compaction during the filtration process [50].

#### 2.3.2. Membrane Optimisation for a Single-Stage Filtration Operation

Batch concentration mode was performed for all the experiments with simultaneous recycling of only the retentate to the feed tank. For each experiment, 3 L of CW were thawed from 4 °C to room temperature (20 °C) and, as a precaution, were sieved with a 0.177 mm mesh to remove suspended particles. This is the maximum operational particle size limitation of the pump, and the sieving process did not retain any solids. In each single-stage filtration using the membrane system, the permeate obtained was collected and measured until the end of the experiment using a balance (WTC 2000, Radwag, PL, USA) and stopwatch. Feed and permeate volumes were measured before and after the filtrations.

To optimize the membrane system in one-stage filtration, the MF 0.2 µm and UF 200 kDa membranes were operated under different operational conditions, namely TMP and CFV. The operating pressure and CFV were maintained similar to that in the pre-conditioning stage (Table 2). Note that the CFV was selected at a maximum of 1.0 m/s due to the system’s limitation.

#### 2.3.3. Membrane Optimisation after CW Pre-Treatment

Once the optimum conditions were determined for each membrane at different CFVs and TMPs, the effect of pre-treatment by lactic acid acidification and centrifugation was individually investigated. Firstly, for lactic acid acidification, 3 L of CW was acidified with 25 mL of lactic acid with 80% of purity (purchased from Labkem, Barcelona, Spain) until a pH of 3 was obtained (measured with pH meter HI1048, Hanna Instruments, Smithfield, RI, USA), and the solution was then stored in the fridge for 24 h for the further separation process. Indeed, the low volume of lactic acid added did not modify the COD concentration of the CW. The choice of lactic acid in the present study was due to three reasons: (i) it is one of the most used coagulants in the dairy industry [31], (ii) its availability, and (iii) its cost.

After 24 h, the CW was sieved with a 0.177 mm mesh. This procedure separated the liquid and any precipitate (Appendix A). For the centrifugation process, 3 L of CW was centrifuged (Thermo-fisher Scientific SL Plus Series Centrifuge, Waltham, MA, USA) at 4200 rpm for 15 min. Further separation of liquid and precipitates was conducted with a 0.177 mm sieve (Appendix A). For comparison purposes, the optimum CFV and TMP from the previous steps were applied to assess the pre-treatment effect on permeate recovery using MF 0.2 µm and UF 200 kDa.

### 2.4. Process Performances

The performance of the membranes was evaluated in terms of permeate flux (J_p_, L/h·m^2^), which was calculated by the ratio between volumetric flow rate (Q, L/h) and membrane-active area (A, m^2^) (Equation (1)):(1)$$J_{p}=\frac{Q}{A}=\frac{V}{t\;A}$$
where V is the permeate volume (L), A is the membrane-active area, and t is the time (h).

The permeate recovery efficiency (Perm_eff_, %) was calculated by considering the volume of the permeate (V_p_, L) compared to the initial volume of the feed (V_f_, L) (Equation (2)):(2)$${\mathrm{Perm}}_{\mathrm{eff}\;\left(\%\right)}=\frac{V_{p}}{V_{f}}\times 100$$

P recovery (P_rec_, %) was calculated based on the mass of P in the permeate compared to that in the feed (Equation (3)):(3)$$P_{\mathrm{rec}\;\left(\%\right)}=\frac{V_{p\;}\;C_{p}}{V_{f\;}\;C_{f}}\times 100$$
where V_p_ (L) represents the final volume of the permeate, C_p_ (g/L) is the concentration of P in the permeate, V_f_ (L) is the initial volume of the feed, and C_f_ is the initial concentration of P in the feed.

### 2.5. Membrane Cleaning

Membrane cleaning was performed by following the recommendations of the manufacturer. After each membrane filtration, the cleaning procedure was conducted, referred to as cleaning in place (CIP). CIP was performed by circulating NaOH 0.05% (*v*/*v*) for 30 min at the TMP and CFV shown in Table 2, followed by a 30-minute soaking period and the final 10-min recirculation prior to flushing with deionized water. After the CIP process, the clean water flux was measured with distilled water at 20 °C under the set pressure level. This procedure aimed to restore the hydraulic membrane permeability and verify the cleaning efficiency of the membrane after each filtration.

The hydraulic permeability (L_p_, L/L/h·m^2^·bar) was calculated based on the ratio of J_p_ and the applied pressure of the system (P, bar) (Equation (4)):(4)$$L_{p}=\frac{J_{p}}{P}=\frac{Q}{\mathrm{AP}}=\frac{V}{\mathrm{tAP}}$$

### 2.6. Freeze Concentration Configuration and Operation

Following filtration, the permeate obtained from the membrane system was then concentrated using PFC. The system has a tank with a 2 L volume (internal diameter of 110 mm and 257.5 mm height), completed with a jacket for thermal control (15 mm thickness) to allow the heat transfer only inside the tank. The chiller (Polyscience AD20R, Niles, IL, USA) was filled with propylene glycol:water (60:40) and was used to circulate the cooling solution through the jacketed tank during the experiment.

The PFC experiment was conducted as follows: 1.8 L of the permeate obtained from the previous membrane filtration was charged to the PFC tank once the refrigerant temperature achieved the selected temperature of −5 °C. The 600 rpm agitation speed (using AM20-D Argo Lab, Carpi, Italy) was applied in the system allowing uniform distribution of the liquid inside the reactor. The selected temperature of −5 °C and stirring speed of 600 rpm were chosen due to the reason that the higher the stirring speed, the higher the mass transfer from the mother solution to the ice fraction; therefore, high solute concentration was accumulated in the concentrated fraction [43,51]. Moreover, the low temperature of −5 °C was chosen because the lower the coolant temperature, the higher the heat transfer between the coolant temperature and the solution, which means that the heat transfer rate is linearly correlated with the temperature difference between the coolant temperature and the solution [37].

A temperature probe was also inserted into the jacked tank’s middle to record the temperature of the concentrated fraction. The refrigerant temperature, the probes, and the room temperature were recorded in the digital data logger connected to the chiller (Figure 2 and Appendix A). To compare the results obtained, 50% of volume reduction was selected in all the experiments. Once the PFC process was completed (achieving 50% volume reduction), the ice fraction (referred as diluted) and concentrated fraction (referred as concentrated) were collected for further analysis.

A mass balance of the P was made to validate the results of the FC experiment, calculated as follows (Equation (5)):(5)$${\mathrm{MB}}_{P}\;\left(\%\right)=\frac{C_{f}V_{f\;}-\left[\left(C_{c}V_{c}\right)+\left(C_{d}V_{d}\right)\right]}{C_{f}V_{f\;}}\times 100$$
where C_in_ (g/L) represents the initial concentration of P in the feeding; C_c_ (g/L) is the concentration of P in the concentrated solution; C_d_ (g/L) is the concentration of P in the ice fraction; V_f_ (L) is the initial volume of the feeding; V_c_ (L) is the volume of the concentrated solution; and V_d_ (L) is the volume of melted ice.

Furthermore, the recovery efficiency (RE_P_) of the P in the concentrated was calculated according to Equation (6) as follows:(6)$${\mathrm{RE}}_{P}\left(\%\right)=\frac{V_{c}C_{c}}{V_{f}C_{f}}\times 100$$

The process efficiency (PE) of the FC process was calculated by the ratio of the concentration of P in the concentrated fraction and the concentration of P in the ice fraction. The PE can be calculated as follows:(7)$$\mathrm{PE}\;\left(\%\right)=\frac{C_{c}-{\;C}_{d}}{C_{c}}\times 100$$

### 2.7. Two-Stage Coupled Technology of Membrane System and FC

A two-stage coupled technology of the membrane system with FC was applied to maximise the P recovery efficiency during CW valorisation and to approach zero-waste disposal. The experimental procedure is illustrated in Appendix A, wherein it is divided into **stage 1** and **stage 2** processes. In the stage 1 process, the CW (previously pre-treated with lactic acid until pH 3) was treated using the membrane system (UF with TMP of 3 bars and CFV of 1 m/s). The permeate obtained from the membrane filtration was further treated with FC, which resulted in the concentrated and diluted fractions. Meanwhile, in the stage 2 process, the retentate obtained from the membrane system in stage 1 was mixed with the diluted fraction obtained from FC treatment in a 50:50 (*v*/*v*) ratio. Afterwards, the mixed solution was treated with the membrane system (UF with TMP of 3 bars and CFV of 1 m/s), and the permeate obtained was further treated with FC. This two-stage operation was conducted in duplicate.

### 2.8. Analytical Methods

The physicochemical characteristics of CW were evaluated including pH, conductivity, total Kjeldahl nitrogen (TKN), chemical oxygen demand (COD), total phosphorus (TP), total solids (TS), volatile solids (VS), and ammonium (N-NH_4_^+^), following the methodologies described in the standard methods for examining water and wastewater [52]. Specifically for TP, 15 mL of the sample was digested for 30 min at 100 °C by adding 5 mL of concentrated sulphuric acid (96%) using a bloc digester (JP Selecta SA, Barcelona, Spain). After 30 min of the first digestion process, 15 mL of nitric acid 3 N was added and digested for 2 h. After allowing the sample to cool, the samples were then filtrated using a 10 µm filter and diluted with deionized water into the measuring range of the analysis. The TP content was measured using the ascorbic acid method [52] and analysed using a spectrophotometer (HACH DR 3900, Loveland, CO, USA) at 880 nm wavelength. Furthermore, the solid concentration of the feed, diluted fraction, and concentrated fraction were measured as ^o^Brix by employing the portable refractometer (model RHB-50 ATC, YHequipment Co. Ltd., Shenzhen City, China) with an accuracy of ±0.1 Brix. The measurement range is between 0 and 50 ^o^Brix.

### 2.9. Energy Consumption

The energy consumption of the coupled application of membrane system and FC was measured separately using a power meter (Maxcio, PM001-GG/CF, Shenzhen City, CH). For the membrane system, the energy consumed was measured considering the pre-condition of the membrane until the filtration had finished as well as the CIP process. For the FC process, the energy consumption was measured considering the pre-cooling of the process (to achieve the set-point temperature) until the freezing process finished. The mechanical agitator was also considered for energy consumption. Finally, the energy consumption of the coupled system was calculated by summating the energy consumed by both systems.

## 3. Results and Discussion

### 3.1. One-Stage Filtration System

#### 3.1.1. Clean Water Flux

The clean water fluxes for each membrane including the cleaning efficiency are presented in Table 3. The results show that the clean water flux measured after the CIP process demonstrated very little difference from the initial value. Interestingly, the cleaning efficiency of the membrane reached 99%, meaning that the CIP successfully removed the cake layers on the membrane surface (reversible fouling) and almost returned the properties of the membrane to the virgin state. It indicated that all membranes used in this study could be reused after each experiment.

#### 3.1.2. MF Performance

The influence of CFV on the performance of the MF membrane is shown in Figure 3. The results show that the higher the CFV applied, the higher the permeate flux rate. For the CFV of 0.5 m/s and TMP of 0.8 bar, the initial permeate flux was 33 L/h·m^2^ and reduced by approximately 38% during the operating time. For the CFV of 0.2 m/s and 0.8 bar of TMP, the initial flux was 22 L/h·m^2^ with 58% permeate flux reduction at the end of the experiment. This result is also supported by that of Atra et al. [53] who reported that applying higher CFV will result in a higher permeate flux rate, meaning that the deposited molecules on the membrane surface are continuously removed and the fouling layers are reduced. The permeate flux rate treating CW was significantly lower, calculated during the steady state condition (min 40–70), which was approximately 89% of the magnitude compared to the average clean water flux (281 L/h·m^2^). This phenomenon was due to absence of fouling during the pre-condition process with deionised water and thus concentration polarisation was minimal [54].

From the beginning of the filtration, a significant permeate flux decline was observed. This was attributed to the rapid formation of the fouling layer, which gradually continued by progressive fouling until the end of the experiment. According to Steinhauer et al. [55], this fouling was mainly due to the high protein and protein aggregated components in whey. For this reason, a higher-pressure force was needed to compensate for the membrane resistance due to the formation of cake fouling layers. Therefore, for the MF in both CFVs and constant TMP, the filtration process had to be stopped to avoid changing the operational conditions. Indeed, as a result, after 150 min of filtration time, 26% of permeate could be recovered by applying a CFV of 0.5 m/s and TMP of 0.8 bar and was used as the set condition for the next experiment.

#### 3.1.3. UF Performance

The performance of the UF membrane is shown in Figure 4. The results indicate that the higher CFV applied to the system, the higher the flux rate obtained, which is consistent with the experiments conducted with the MF membrane and agrees with the literature [53,56]. An increase of CFV in the system generates higher turbulence, scours away the accumulated cake deposition, and reduces the overall membrane hydraulic resistance. For the CFV of 1.0 m/s and TMP of 2 bars, the initial flux was 28 L/h·m^2^ with 30% of the flux reduction at the end of the experiment, while the CFV of 0.5 m/s and TMP of 2 bars gave an initial flux of 22 L/h·m^2^ and a reduction of 48%. The lowest CFV applied to the system was 0.2 m/s with a constant TMP of 2 bars. This resulted in the lowest initial flux rate of 11 L/h·m^2^ with 72% flux reduction. The UF membrane exhibited a lower permeate flux rate when compared to the MF membrane; this is due to the UF membrane having a smaller pore size compared to the MF membrane, which corresponds to lower permeability and hydraulic performances [57].

At the TMP of 2 bars, the last result showed that a CFV of 1.0 m/s yields the highest permeate rate. Then, the effects of different TMPs (2 bars and 3 bars) with a constant CFV (1.0 m/s) were also examined. The results revealed that, as shown in Figure 4, the TMP of 3 bars with a constant CFV of 1 m/s resulted in a higher permeate flux rate. The TMP generates an increase in the permeate flux, which also aligned with the findings of Sofuwani et al. [58] and Cassano et al. [59]. By applying a TMP of 3 bars, the initial flux reached was 36 L/h·m^2^ with a flux reduction of 36% at the end of the experiment. While for a TMP of 2 bars, the initial flux was 27 L/h·m^2^ and the flux reduction was 11%. However, there is a critical point, later called critical flux, in which this linear relationship is neglected, meaning that the maximum flux obtained cannot be increased by increasing TMP. Accordingly, from minute 100 to 150, the permeate flux rate in both TMPs (2 bars and 3 bars) under a constant CFV of 1.0 m/s were close. Therefore, the selected TMP should be lower than the critical zone in which fouling effects are also minimum [59]. Finally, according to Figure 4, a CFV of 1.0 m/s with a TMP of 3 bars are considered the optimum operational conditions for the application of the UF membrane with 30% of permeate recovery.

### 3.2. Effects of Pre-Treatments on Permeate and P Recovery for MF and UF Membranes

A pre-treatment of CW by lactic acid acidification to pH 3 and clarification by centrifugation was individually assessed to increase the permeate recovery from the membrane process. The performances of MF and UF membranes yielded low permeate recoveries as described previously, which were up to 26% and 30%, respectively. The pre-treatment by lactic acid to pH 3, as shown in Figure 5, successfully increased the permeate flux rate up to 50% for MF and 68% for UF compared to those without the treatment process (calculated based on Equation (2)). This is due to the lactic acid acidification to pH 3 improves the protein’s solubility and avoid the insolubilisation of calcium and calcium phosphate. The initial pH of CW was 4.46 (Table 1), which corresponds to the isoelectric point of whey protein or protein aggregates [31], where electrostatic interactions between protein and protein aggregates are powerless. Therefore, the protein deposition in the membrane surface is enhanced. Thus, pH 3 is proven to effectively increase the permeate flux rate due to whey’s electrostatic point, which was also agreed with the study of Konrad et al. [31]. After the acidification process, the precipitates could be separated from their liquids; however, in this study, the composition of the precipitates of the CW was not analysed. According to Mourouzidis-Mourouzis and Karabelas [60], the precipitates are mainly composed of β-Lg and α-La, which are responsible for accelerating membrane fouling during whey filtration.

Besides the lactic acid acidification, the whey clearance using centrifugation was also tested. After the centrifugation process, a precipitate was formed in the bottom of the tubes. In this study, the composition of the precipitates was not analysed; however, according to Steinhauer et al. [55], the precipitates might be calcium and/or calcium phosphate, which are also involved during the filtration of CW and lead to pore blockage and severe fouling [35,61]. Figure 5 shows that the effects of the centrifugation process were less pronounced than that of the lactic acid acidification and could only recover 46% of the permeate for MF and 59% for UF during the filtration. The cake accumulation was slower to accumulate compared to without treatment; this can be proven by the longer filtration time, which was 360 min with a constant TMP. It indicates that the pre-treatment processes lead to better long-term membrane performances and fewer membrane fouling phenomena.

The observation was made that the pre-treatment as lactic acid acidification did not change the P composition in the solution. During the separation process, the P was mostly (99%) found in the supernatant and not in the precipitates. At the end of the filtration process (at min 360), both retentate and permeate were collected for further P analysis. The results showed that with lactic acid acidification (considered the best pre-treatment), the P was mainly passing the membrane surface, meaning that P was recovered more in the permeate (55%) for the MF. Similarly, for the UF, the P was found at 66% in the permeate, while 34% was in the retentate. This phenomenon occurred due to the particle size of the PO_4_^3−^ (0.46 nm) being lower than the MF and UF membrane pore sizes, which were 0.2 µm and 200 kDa, respectively [7,62]. Among all the experiments performed, the application of the UF membrane with lactic acid acidification to pH 3 as a pre-treatment was identified as the most promising treatment for the valorisation of CW based on the permeate recovery and the P concentration in the permeate. Consequently, the UF membrane was selected for the application of the membrane filtration system coupled with FC.

### 3.3. Two-Stage Coupled Technology of Membrane System and FC for P Recovery

Under the optimal operational conditions obtained using membrane system (UF with lactic acid acidification to pH 3 under CFV 1 m/s and TMP 3 bars), the recovered permeate was subsequently treated with FC to obtain a P-rich concentrate. According to Figure 6, the results indicate that in the stage 1 process, 40% of P could be recovered from the initial CW with 77% of process efficiency. Meanwhile, in stage 2 of the two-stage process, a further 30% of P could be recovered from the initial CW. Thus, by applying a two-stage coupled technology of membrane system and FC, 70% of P could be recovered. Furthermore, 50% of water can be recovered (diluted fraction from FC) as reclaimed water, which must be analysed in order to evaluate its potential to be reused for membrane cleaning or for agricultural irrigation according to the Spanish and European regulation. Indeed, according to Figure 6, the ion selectivity and P specification could comprehensively be understood by the coupling membrane system coupled to FC compared to applying only FC.

Meanwhile, the retentate obtained from stage 2 of the membrane system was collected and could be further valorised. According to Pires et al. [63], the retentate is rich with whey protein, fats, insoluble salts, lactose, and soluble minerals. Some studies were attempted to valorise the retentate, for example, Onwulata & Huht [64] employed UF retentate coupled with diafiltration process to produce whey protein concentrate, whey protein isolates, and whey protein hydrolysates. Furthermore, Henriques et al. [65] attempted to use UF retentate coupled with freeze drying as one of the ingredients in yoghurt formulation, while Hinrichs [66] employed the UF retentate coupled with heating process to produce standardised cheese milk.

From this study, the electricity required to recover P from CW by employing the coupled technology of membrane system and FC was 2.9 kWh in the first stage (40% P recovery). Meanwhile, to recover 70% of P from CW, the electricity consumed was double, which was approximately 5.8 kWh. It can be translated that, in the two-stage operation, the electricity needed was 8300 kWh/kg P recovered. This electricity consumption was higher compared to that of other technologies, for example, electrochemical phosphorus recovery at the lab scale treating domestic wastewater consumed electricity of up to 2238 kWh/kg P recovered [67]. Indeed, the energy consumed mainly depends on the P concentration in the feed stream, the technology configuration (single or coupled system), the power rating of the equipment, and the targeted recovery rate [68]. Furthermore, this study and the calculation were based on the lab-scale application; therefore, the electricity consumption will be cheaper when applied on an industrial scale due to the economics of scale.

## 4. Conclusions

The application of a novel hybrid membrane process coupled with freeze concentration for P recovery from CW has shown promising results. The optimum conditions obtained from this study were from using a UF 200 kDa membrane with a CFV of 1 m/s and TMP of 3 bars with lactic acid acidification as a pre-treatment to the membrane system. Further coupling with FC, the recovery efficiency of P achieved up to 40% from the initial CW. A two-stage coupled system could be suggested for larger-scale applications since the P recovery resulted in a higher P recovery efficiency (70%), obtaining a P-rich product, and can potentially applied as an ingredient for the formulation of tailor-made fertilisers. Furthermore, the potential for water recycling from FC and whey protein recovery from the UF retentate could be obtained, which could lead to better process economics and zero liquid discharge.

In the end, drawing from the present study, this novel and innovative coupled technology for P recovery contributes to ensuring long-term availability and accessibility of sustainable P supply, minimising the environmental impacts associated with excess P, and possibly replacing the dependency of imported PR and constituting a further step toward better sustainability and the establishment of a broader circular economy framework.

## Figures and Tables

**Figure 1 membranes-13-00450-f001:**
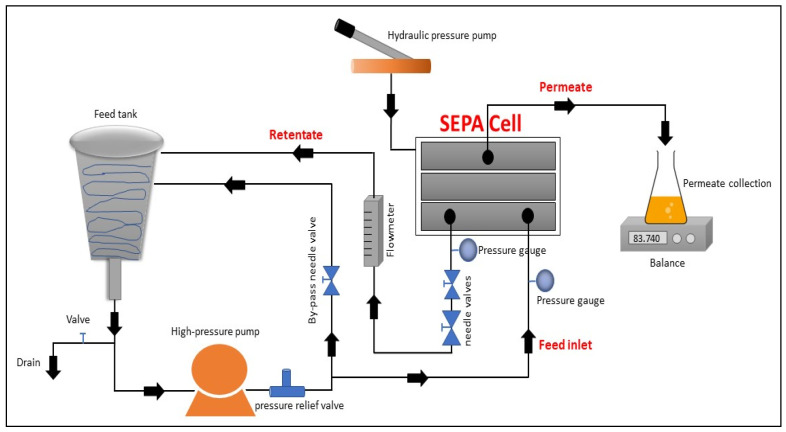
The schematic diagram of the membrane filtration apparatus.

**Figure 2 membranes-13-00450-f002:**
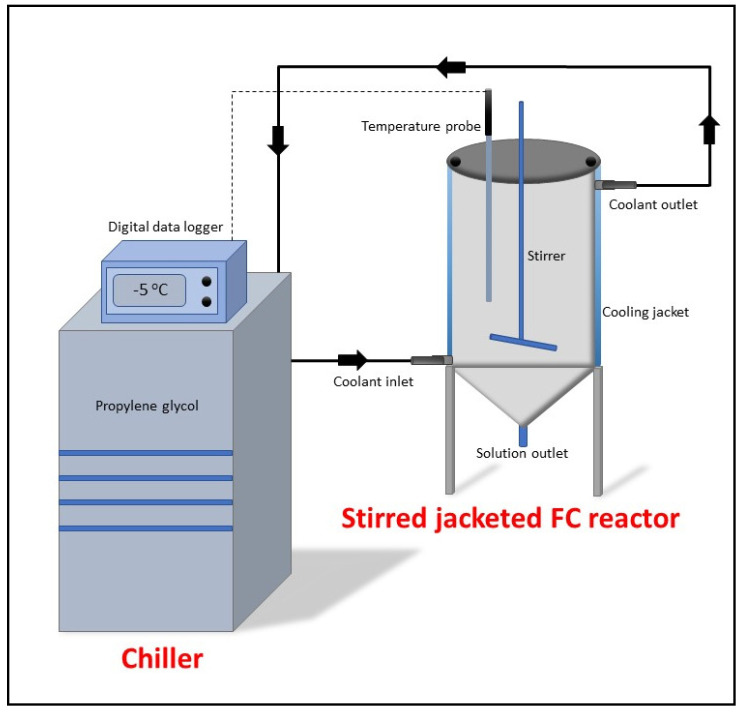
Schematic design of the lab scale progressive freeze concentration system.

**Figure 3 membranes-13-00450-f003:**
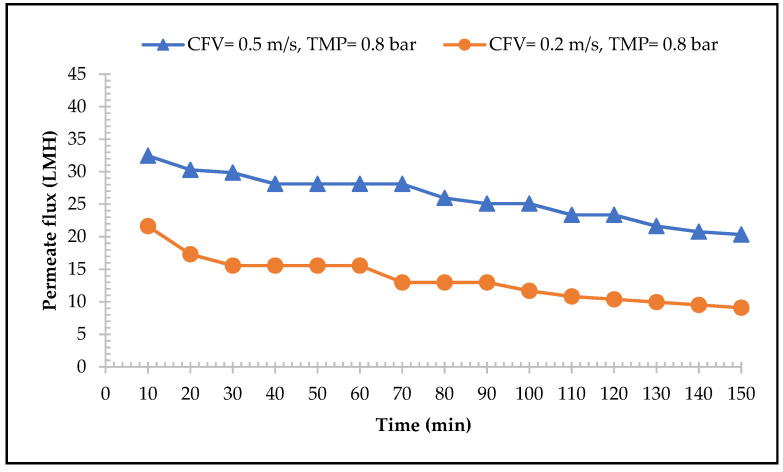
The permeate flux of 0.2 µm MF membranes with the function of time with different CFVs and constant TMP.

**Figure 4 membranes-13-00450-f004:**
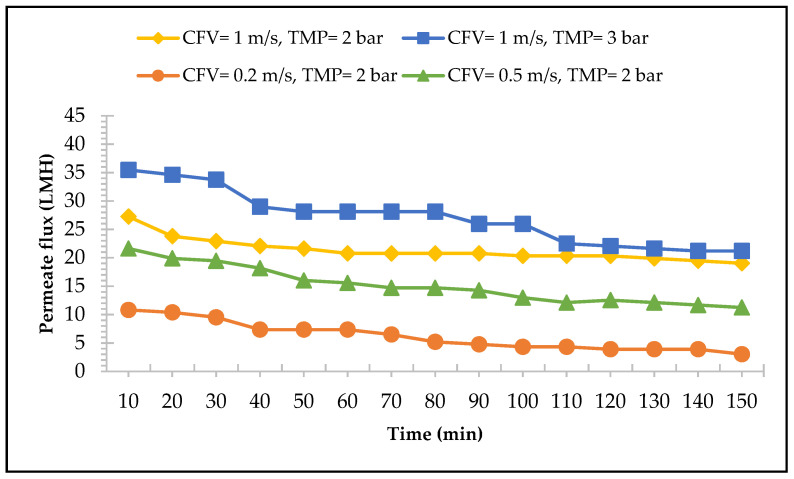
The permeate flux of UF membrane with the function of time in different CFVs and TMPs.

**Figure 5 membranes-13-00450-f005:**
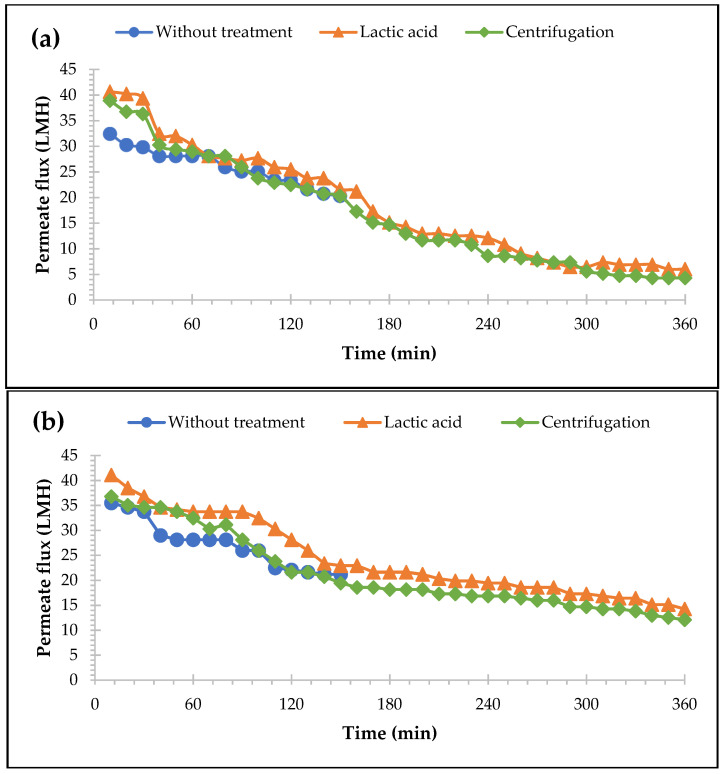
Effects of pre-treatments on the permeate flux of (**a**) MF membranes 0.2 µm (CFV = 0.5 m/s and TMP = 0.8 bar) and (**b**) UF membranes 200 kDa (CFV = 1.0 m/s and TMP = 3 bar) with the function of time.

**Figure 6 membranes-13-00450-f006:**
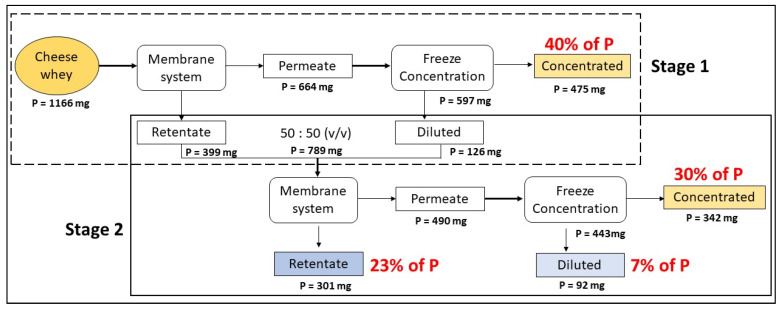
Phosphorus recovery percentage obtained during stages one and two of membrane filtration system coupled with freeze concentration.

**Table 1 membranes-13-00450-t001:** The physicochemical characterization of CW.

Parameter	Value	Unit
Total solids (TS)	69 ± 1	g/kg
Volatile solids (VS)	59 ± 1	g/kg
Total Kjeldahl nitrogen (TKN)	0.80 ± 0.03	g/L
N-ammonium (N-NH4+)	0.37 ± 0.07	g/L
Total phosphorus (TP)	0.39 ± 0.03	g/L
Chemical oxygen demand (COD)	87 ± 1	g/L
Conductivity	8.36 ± 0.05	mS/cm
pH	4.46 ± 0.02	-

**Table 2 membranes-13-00450-t002:** Pre-conditions of the membrane system.

Membrane Module	CFV (m/s)	TMP (bar)
MF 0.2 µm	0.2	0.8
	0.5	0.8
UF 200 kDa	0.2	2
	0.5	2
	1.0	2
	1.0	3

**Table 3 membranes-13-00450-t003:** Clean water fluxes and hydraulic permeabilities (Lp) of each membrane using distilled water at 25 °C, including the cleaning efficiency.

Membrane	Clean Water Flux (L/h·m^2^)	Water Hydraulic Permeability (L/h·m^2^·bar)	Pressure (bar)	CFV (m/s)	Cleaning Efficiency (%)
MF 0.2 µm	281 ± 2	351.5 ± 0.2	0.8	0.2 and 0.5	99
UF 200 kDa	230 ± 1	76.4 ± 0.5	3	0.2, 0.5, and 1	99
UF 200 kDa	230 ± 1	115.2 ± 0.1	2	0.2, 0.5, and 1	99

## Data Availability

Not applicable.

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
