# Peer review of "A Novel Hybrid Membrane Process Coupled with Freeze Concentration for Phosphorus Recovery from Cheese Whey"

_membranes, 2023, doi:10.3390/membranes13040450_

Round 1

Reviewer 1 Report

The manuscript investigated a novel hybrid membrane process coupled with freeze concentration for phosphorus recovery from cheese whey. It gave an alternative approach to recover phosphorus in wastewater with high content of organic matter. Some comments are provided for potential improvement of the manuscript.

1. Why did the author use MF and UF before freeze concentration? What would it be if freeze concentration was directly applied to the raw cheese whey without MF and UF?

2. Line 77, why did the author use lactic acid for the acidification? It was unclear how much amount was used. Maybe it was not the best choice since the addition of lactic acid might increase COD of the wastewater.

3. Table 2, only 2 sets of parameters were investigated in MF performance, therefore, the so-called “optimum operational conditions” was in doubt.

4. Fig. 4, the optimum condition was obtained at the highest CFV and TMP, thus, it was unclear whether the performance will be better if higher CFV and TMP were used.

5. Fig. 5, the difference of the permeate fluxes among the three sets of experiments were not obvious, but the permeate recovery was much higher in the group of “Lactic acid” as compared to “Without treatment” as claimed in lines 351-353.

6. Line 359, it was unreasonable to claim that “pH 3 is considered the optimal pH for filtration due to under whey’s electrostatic point” since no other pH values were compared in this study. For example, pH of 2 or 4 are all lower than the electrostatic point of 4.46, and their effects were unknown.

7. Freeze concentration was an important technology in P recovery process, but there were no investigations on the freeze concentration. If freeze concentration for P recovery had been widely studied, references should be added.

Author Response

Thank you very much for the opportunity to revise our manuscript. We appreciate the time and effort that the reviewer dedicated to providing feedback on our manuscript and are grateful for the insightful comments and valuable improvements to our paper. We have incorporated most of the suggestions made by reviewer and we think that the paper has improved, and it is ready for publication. Please find a summary of our responses to the major points that the reviewer pointed out.

  1. Why did the author use MF and UF before freeze concentration? What would it be if freeze concentration was directly applied to the raw cheese whey without MF and UF?

Thank you for reaching out this comment. It really depends on the objectives of the study. Firstly, the novelty of this study was to evaluate the phosphorus recovery efficiency from CW by coupling membranes system (MF and UF) with freeze concentration technology. Furthermore, by applying the MF and UF before freeze concentration, it helps us to understand the ion selectivity and speciation during the filtration process, especially phosphorus. Additionally, the study of freeze concentration treating cheese whey has been revealed, for example in the studies of Canella et al., (2018), Sánchez  et al., (2011) and Belén  et al., (2018). A new reference (Uald Lamkaddam et al. (2023)) has been added to strengthen the literature about the direct application of freeze concentration for CW treatment and a sentence has been added in the manuscript:

Lines 94-97: “Furthermore, the study of Uald-Lamkaddam et al. [69] applied progressive FC to recover protein and lactose from CW. The results showed that 13.9% (w/w) recovered in the concentrated fraction, meanwhile 6.2% (w/w) of lactose could be recovered in the ice fraction.”

To emphasize the advantage of using the coupled technology of membrane system and FC compared to only FC, a sentence has been incorporated in the manuscript:

Lines 419-421: “Indeed, according to Figure 6, the ion selectivity and P specification could comprehensively be understood by coupling membrane system coupled to FC compared to applying only FC”

  1. Line 77, why did the author use lactic acid for the acidification? It was unclear how much amount was used. Maybe it was not the best choice since the addition of lactic acid might increase COD of the wastewater.

Thank you for pointing out this comment. The selection of lactic acid for the acidification was due to 3 reasons: i) Lactic acid is the most widely used coagulant in the dairy industry, ii) Chemical availability, and iii) Cost. Furthermore, 25 mL of lactic acid were added to 3 L of raw CW for the acidification. Indeed, with low volume of lactic acid over the volume of raw CW, the COD concentration after acidification was not significantly changed. In this case, the average COD of raw CW and after acidification were 87 g/L and 88.5 g/L, respectively. For the clarifications, a sentence indicating this reason has been incorporated in the Material and Methods section as follows:

Firstly, for lactic acid acidification, 3 L of CW were acidified with 25 mL of lactic acid with 80% of purity (purchased from Labkem, Barcelona, Spain) until pH 3 was obtained (measured with pH meter HI1048, Hanna Instruments, USA) and the solution was then stored in the fridge for 24 h for further separation process. Indeed, the low volume of lactic acid added, it did not modify the COD concentration of the CW. The choice of lactic acid in the present study was due to 3 reasons: i) it is one of the most used coagulants in the dairy industry [31], ii) its availability and iii) cost.

  1. Table 2, only 2 sets of parameters were investigated in MF performance, therefore, the so-called “optimum operational conditions” was in doubt.

Considering the reviewer´s comment, the manuscript has been revised and changes have been included in the manuscript as shown below:

“Indeed, as a result, after 150 min of filtration time, 26% of permeate could be recovered by applying CFV of 0.5 m/s and TMP of 0.8 bar and used as the set condition for the next experiment.”

  1. 4, the optimum condition was obtained at the highest CFV and TMP, thus, it was unclear whether the performance will be better if higher CFV and TMP were used.

In this present study, for the UF membrane (Figure 4), we evaluated different ranges of CFV (0.2, 0.5 and 1 m/s) and TMP (2 and 3 bar). The results showed that, firstly, in constant TMP of 2 bar, the highest permeate flux was obtained at 1 m/s of CFV. Then, the test has been made in constant CFV of 1 m/s and TMP of 2 and 3 bars. In this condition, the higher permeate flux was obtained at the higher TMP of 3 bar. For the clarification, a modification has been added in the manuscript as follows:

“The results revealed that, as shown in Figure 4, the TMP of 3 bar with constant CFV of 1 m/s resulted in the higher permeate flux rate.”

  1. 5, the difference of the permeate fluxes among the three sets of experiments were not obvious, but the permeate recovery was much higher in the group of “Lactic acid” as compared to “Without treatment” as claimed in lines 351-353.

Thank you very much for bringing up this comment. Indeed, the difference of the permeate fluxes were not significant. However, the permeate recovery rate (see equation 2) of the lactic acid group was contrast and the highest among others. This is due to the longer filtration time.

  1. Line 359, it was unreasonable to claim that “pH 3 is considered the optimal pH for filtration due to under whey’s electrostatic point” since no other pH values were compared in this study. For example, pH of 2 or 4 are all lower than the electrostatic point of 4.46, and their effects were unknown.

In the present study, pH 3 was chosen due to pH 3 improves the protein’s solubility and avoid the insolubilisation of calcium and calcium phosphate. Furthermore, the pH 3 was also lower than the electrostatic point of whey protein or protein aggregates which was also agreed with the study of Konrad et al., 2012. Furthermore, the previous affirmation written as an optimal pH for filtration has been modified and changes have been included in the manuscript as follows:

“Thus, pH 3 is proven to effectively increase the permeate flux rate considered the optimal pH for filtration due to under whey’s electrostatic point which was also agreed with the study of Konrad et al. [31].”

  1. Freeze concentration was an important technology in P recovery process, but there were no investigations on the freeze concentration. If freeze concentration for P recovery had been widely studied, references should be added.

Thank you for pointing out this comment. Indeed, from our knowledge, there were no study investigating the only P recovery using freeze concentration specifically from cheese whey. The use of freeze concentration for nutrients recovery (including P) was investigated using different feedstock, for example Uald-lamkaddam et al. (2021) using nutrient-rich effluent generated from the anaerobic digestion of agro-industrial waste as a feedstock.

Uald-lamkaddam, I., Dadrasnia, A., Llenas, L., Pons, S., Col, J., Vega, E., & Mora, M. (2021). Application of Freeze Concentration Technologies to Valorize Nutrient-Rich Effluents Generated from the Anaerobic Digestion of Agro-Industrial Wastes.

Reviewer 2 Report

The study is interesting and well presented.

I have some comments that should be addressed before publication:

CVF should be corrected to CFV in Table 3

When discussing the permeate flux compared to clean water flux you state that “the permeate flux rate treating CW was significantly lower and was approximately 89% of the magnitude” (line 299-300). This can be understood as 89% of the clean water flux, but I believe 89% reduction compared to the clean water flux is meant. This point should be clarified and also if the comparison is made for the initial flux or after some specific time.

The y-axis of Figure 4 should start at 0, not -5

It is written: “The pre-treatment lactic acid to pH 3, as shown in Figure 5, was successfully increased the permeate flux rate up to 50% for MF and 68% for UF compared to without the treatment process.” (line 351-354) => I don’t see this in Figure 5, please explain in more detail how you have reached this results and which data points that are used in the comparison.

It is written: “The cake accumulation was slower to accumulate compared to without treatment; this can be proven by the longer filtration time which was 360 min. It indicates that the pre-treatment processes lead to better long term- membrane performances and fewer membrane fouling phenomena.” => It is clear that the filtration time was longer in the experiments with pre-treatment, but the permeate flux curves without pre-treatment and with pre-treatment are similar after 150 minutes so it is unclear why the experiments without pre-treatment could not be run longer and thus achieved a higher permeate recovery

Author Response

Thank you very much for the opportunity to revise our manuscript. We appreciate the time and effort that the reviewer dedicated to providing feedback on our manuscript and are grateful for the insightful comments and valuable improvements to our paper. We have incorporated most of the suggestions made by reviewer and we think that the paper has improved, and it is ready for publication. Please find a summary of our responses to the major points that the reviewer pointed out.

  1. CVF should be corrected to CFV in Table 3

Thank you very much for your kind correction. The manuscript has been revised and we have made the modification accordingly with track changes option.

  1. When discussing the permeate flux compared to clean water flux you state that “the permeate flux rate treating CW was significantly lower and was approximately 89% of the magnitude” (line 299-300). This can be understood as 89% of the clean water flux, but I believe 89% reduction compared to the clean water flux is meant. This point should be clarified and also if the comparison is made for the initial flux or after some specific time.

Indeed, the 89% reduction was the comparison between permeate flux treating CW over the clean water flux as included in the manuscript.  Therefore, for the clarification, the changes have been incuded in the manuscript as shown below:

“The permeate flux rate treating CW was significantly lower, calculated during the steady state condition (min 40 – 70), which was approximately 89% of the magnitude compared to the average clean water flux (281 L/h·m2).

  1. The y-axis of Figure 4 should start at 0, not -5

As suggested by the reviewer, the manuscript has been revised and we have made the modification accordingly.

  1. It is written: “The pre-treatment lactic acid to pH 3, as shown in Figure 5, was successfully increased the permeate flux rate up to 50% for MF and 68% for UF compared to without the treatment process.” (line 351-354) => I don’t see this in Figure 5, please explain in more detail how you have reached this results and which data points that are used in the comparison.

Thank you for pointing this out. The calculation was made based on the total volume of permeate collected during the filtration time (duration of the cycle: 360 min for lactic acid acidification and 150 min without pre-treatment) divided by the initial volume of the cheese whey. In this case, please refers to the equation 2. For the clarifications, the changes are included in the manuscript as shown below:

“The pre-treatment as lactic acid to pH 3, as shown in Figure 5, successfully increased the permeate flux rate up to 50% for MF and 68% for UF compared to without the treatment process (calculated based on Equation 2).

  1. It is written: “The cake accumulation was slower to accumulate compared to without treatment; this can be proven by the longer filtration time which was 360 min. It indicates that the pre-treatment processes lead to better long term- membrane performances and fewer membrane fouling phenomena.” => It is clear that the filtration time was longer in the experiments with pre-treatment, but the permeate flux curves without pre-treatment and with pre-treatment are similar after 150 minutes so it is unclear why the experiments without pre-treatment could not be run longer and thus achieved a higher permeate recovery.

Thank you for pointing this out. As stated in the manuscript, due to the cake accumulation on the membrane surface, the transmembrane pressure increased from the set point especially after min 150 of filtration and the experiment had to be stopped to avoid changing the operational conditions. For the clarifications, the changes are included in the manuscript as shown below:

“The cake accumulation was slower to accumulate compared to without treatment; this can be proven by the longer filtration time which was 360 min with a constant TMP. It indicates that the pre-treatment processes lead to better long term- membrane performances and fewer membrane fouling phenomena.”

Reviewer 3 Report

This article nicely describes a novel hybrid membrane process coupled with freeze concentration for phosphorus recovery from Cheese whey. The authors nicely present a series of comprehensive experimental to assess the technical feasibility of phosphorus recovery from Cheese whey. The authors have used a novel and innovative coupled technology for phosphorus recovery. Due to the increasing demand for phosphorus fertilizer, recovery of phosphorus is one of the emerging problems in the world. I think this work is an immersive study in the field of selectively extracting various ions of high value and is within the scope of the Journal. I think authors have presented the scientific data in an excellent manner. Therefore, I recommend publication in its present form. Though authors need to revise the whole manuscript e.g., crosscheck some spelling mistakes and grammatical errors through out the manuscripts before publication.

Author Response

Thank you very much for the opportunity to revise our manuscript. We appreciate the time and effort that the reviewer dedicated to providing feedback on our manuscript and are grateful for the insightful comments and valuable improvements to our paper. We have incorporated most of the suggestions made by reviewer and we think that the paper has improved, and it is ready for publication. Please find a summary of our responses to the major points that the reviewer pointed out.

  1. This article nicely describes a novel hybrid membrane process coupled with freeze concentration for phosphorus recovery from Cheese whey. The authors nicely present a series of comprehensive experimental to assess the technical feasibility of phosphorus recovery from Cheese whey. The authors have used a novel and innovative coupled technology for phosphorus recovery. Due to the increasing demand for phosphorus fertilizer, recovery of phosphorus is one of the emerging problems in the world. I think this work is an immersive study in the field of selectively extracting various ions of high value and is within the scope of the Journal. I think authors have presented the scientific data in an excellent manner. Therefore, I recommend publication in its present form. Though authors need to revise the whole manuscript e.g., crosscheck some spelling mistakes and grammatical errors throughout the manuscripts before publication.

Special thanks to you for your good comments. We went through the entire manuscript to eliminate spelling mistakes and grammatical errors and we have made revisions accordingly.